# Machine Learning Models for Tracking Blood Loss and Resuscitation in a Hemorrhagic Shock Swine Injury Model

**DOI:** 10.3390/bioengineering11111075

**Published:** 2024-10-27

**Authors:** Jose M. Gonzalez, Ryan Ortiz, Lawrence Holland, Austin Ruiz, Evan Ross, Eric J. Snider

**Affiliations:** Organ Support and Automation Technologies Group, U.S. Army Institute of Surgical Research, Joint Base San Antonio, Fort Sam Houston, San Antonio, TX 78234, USA; jose.m.gonzalez355.civ@health.mil (J.M.G.);

**Keywords:** feature extraction, hemorrhage, machine learning, deep learning, predictive modeling, advanced monitoring, swine, shock

## Abstract

Hemorrhage leading to life-threatening shock is a common and critical problem in both civilian and military medicine. Due to complex physiological compensatory mechanisms, traditional vital signs may fail to detect patients’ impending hemorrhagic shock in a timely manner when life-saving interventions are still viable. To address this shortcoming of traditional vital signs in detecting hemorrhagic shock, we have attempted to identify metrics that can predict blood loss. We have previously combined feature extraction and machine learning methodologies applied to arterial waveform analysis to develop advanced metrics that have enabled the early and accurate detection of impending shock in a canine model of hemorrhage, including metrics that estimate blood loss such as the Blood Loss Volume Metric, the Percent Estimated Blood Loss metric, and the Hemorrhage Area metric. Importantly, these metrics were able to identify impending shock well before traditional vital signs, such as blood pressure, were altered enough to identify shock. Here, we apply these advanced metrics developed using data from a canine model to data collected from a swine model of controlled hemorrhage as an interim step towards showing their relevance to human medicine. Based on the performance of these advanced metrics, we conclude that the framework for developing these metrics in the previous canine model remains applicable when applied to a swine model and results in accurate performance in these advanced metrics. The success of these advanced metrics in swine, which share physiological similarities to humans, shows promise in developing advanced blood loss metrics for humans, which would result in increased positive casualty outcomes due to hemorrhage in civilian and military medicine.

## 1. Introduction

### 1.1. Motivation

Hemorrhagic shock is a significant and urgent concern in civilian and military trauma care, where it is the leading cause of preventable death in both the civilian healthcare system and on the battlefield [1,2]. In recent conflicts, such as the ongoing Russo-Ukrainian war, severe blood loss has been a common and devastating injury among soldiers, with nearly 45% of casualties treated at field medical facilities suffering from hemorrhage [3]. Traditional methods for detecting shock, relying on clinical signs such as low blood pressure (hypotension), rapid heart rate (tachycardia), and altered mental status [4], are typically inadequate because these signs only become evident after a substantial amount of blood has already been lost, delaying diagnosis and critically impacting the timely delivery of life-saving interventions.

The pressing need for earlier and more accurate detection methods remains a critical focus for biomedical research. Researchers are increasingly focusing on identifying physiological and biological markers that could indicate the onset of hemorrhagic shock well before traditional symptoms appear [5,6,7]. These markers have the potential to revolutionize trauma care by allowing for much earlier diagnosis and intervention, thereby improving outcomes for casualties involving hemorrhagic shock. Moreover, the complexity of modern warfare and pre-hospital care in which military and civilian personnel operate further complicates the timely and accurate detection of hemorrhagic shock. Medical working conditions are often harsh and chaotic, making reliance on traditional diagnostic methods even less practical. Therefore, there is a critical need to develop robust, portable, and easy-to-use diagnostic tools that can function effectively in these challenging settings.

These challenges underscore the need for innovative approaches like machine learning, which can analyze complex physiological data in real time to detect earlier signs of hemorrhagic shock. By applying machine learning algorithms to features extracted from physiological signals, we aim to identify subtle changes that are not immediately visible through traditional diagnostic methods, bridging the gap between biological data and actionable clinical insights. By leveraging these techniques, we hope to develop more sensitive and earlier indicators of hemorrhagic shock, which can be integrated into practical diagnostic tools suitable for use in the field. Our goal is to enhance the ability of medical personnel to provide timely and effective care, thereby reducing mortality rates and improving outcomes for patients suffering from severe blood loss.

### 1.2. Overview of Feature Extraction

Feature extraction is a signal processing method that identifies characteristics or patterns hidden in an input signal and is often synonymous with an input variable used for artificial intelligence (AI) algorithms [8]. Features can result from a variety of mathematical manipulations done to the signal; for example, features can be absolute magnitudes, differences between features, time-domain manipulations, frequency-domain manipulations, algebraic manipulations of multiple features, as well as many other methodologies for producing features [9]. Features extracted from a signal minimize the amount of data required as an input to an algorithm, as fewer features can represent a large amount of data points in a raw signal while still maintaining original information and sufficient accuracy, which is necessary to develop efficient and robust AI algorithms [9,10].

### 1.3. Overview of Machine Learning and Deep Learning

Machine learning (ML) and deep learning (DL) are subsets of AI that focus on building algorithms capable of learning and making predictions based on data. ML involves training models to find patterns in data and make decisions. DL is a specialized branch of ML that utilizes neural networks composed of a hierarchy of nodes. The nodes that compute data are organized into layers that feed the computational outputs of data into the next, deeper layer, which analyzes more specific patterns in data based on the output of the previous layer. Advancements in AI have the potential to revolutionize biomedical research by providing advanced tools for data analysis and interpretation. Possible applications include medical imaging, predictive analysis, wearable devices, and genomics [11,12]. In the context of shock detection, ML and DL methods offer the potential to develop robust predictive models by extracting relevant features from physiological signals and identifying early indicators of hemorrhagic shock. By leveraging these technologies, we can improve the accuracy and timeliness of shock detection, ultimately enhancing patient outcomes.

### 1.4. Previous Work

Septic shock, the final and life-threatening stage of sepsis, is a dramatic drop in blood pressure that can damage vital organs and result in death. Early treatment of sepsis is critical for the survival of a patient. Previous groups have used ML to predict septic shock with earlier detection times compared to previously developed models, opening the pathway for earlier treatment and increasing survivability of sepsis [13]. Hemodynamic shock, which is a failure of the circulatory system resulting in severe organ failure or death, is often a result of significant blood loss. Unfortunately, symptoms of hypovolemic shock often become apparent when life-saving interventions are less effective [14]. Previous groups have successfully used thermal images to predict shock at intervals where time is equal to zero, three, six, and twelve hours, which would allow life-saving interventions to occur at points where they are more effective in increasing patient survivability [15]. We have successfully explored several novel methods for early shock detection. These methods include using arterial line waveform to calculate Blood Loss Volume Metric (BLVM), Percent Estimated Blood Loss (PEBL), and Hemorrhage Area (HemArea) in a controlled canine hemorrhage model [16,17], and compensatory reserve measurement (CRM) in a simulated human hemorrhage model [18]. In both studies, features were extracted from an arterial waveform signal, and a minimal-redundancy maximal-relevance (MRMR) criterion was used to rank the features against the desired metric. Using the top features as input to the ML models, the models were trained to predict each desired metric of the respective studies.

### 1.5. Study Aim

The BLVM, PEBL, and HemArea metrics showed promising potential in identifying the onset of shock in canine models before traditional signs associated with hemorrhagic shock began to appear. However, these efforts were aimed at identifying whether advanced hemorrhage detection metrics were suitable for military working dogs. In this study, we aim to evaluate the effectiveness of our previously established early shock detection methods developed for canines in a swine model subjected to a controlled hemorrhage. Swine are widely regarded as an ideal preclinical model for human physiology due to their comparable cardiovascular and metabolic responses to hemorrhage [19]. By validating these ML model development techniques in a species with similar physiology to humans, we seek to strengthen the evidence for their potential application in human medicine. This research is crucial for developing reliable triage tools that can be used in both clinical and combat field settings to promptly identify and treat hemorrhagic shock, ultimately saving lives.

## 2. Materials and Methods

### 2.1. Hemorrhagic Shock Swine Injury Model

Datasets captured in a previously performed swine (*Sus scrofa domestica*) hemorrhagic shock injury and fluid resuscitation model were used for training and developing the ML models in this research effort [17]. Research was conducted in compliance with the Animal Welfare Act, implementing Animal Welfare regulations and the principles of the Guide for the Care and Use of Laboratory Animals. This study was approved by the Institutional Animal Care and Use Committee (IACUC). The facility where this research was conducted is fully accredited by AAALAC International.

Briefly, animals were maintained under a surgical plane of anesthesia using 0–5% isoflurane titrated to effect. Analgesia was provided throughout the study with buprenorphine SR. In this study, swine subjects were first instrumented with femoral catheters for controlled hemorrhage (artery) and resuscitation (vein). A carotid artery catheter was placed for arterial pressure readings (Arrow International, Morrisville, NC, USA), and an 8.5 Fr. percutaneous sheath introducer was placed through which a pulmonary artery Swan-Ganz catheter (Edwards Lifesciences, Irvine, CA, USA) was advanced into the pulmonary artery for cardiac output monitoring. Next, an open splenectomy was performed, followed by a 30-min stabilization period.

Then, a controlled hemorrhage to a target mean arterial pressure (MAP) of 35 mmHg was performed to induce hypovolemic shock, wherein an automated hemorrhage decision table (AutoBleed) controlled the rates of blood removal to reach this MAP target. Removed blood was immediately mixed with CPDA-1 solution (citrate, phosphate, dextrose, adenine at a 1:7 volumetric ratio) for anticoagulation. Animals were held at this pressure target under AutoBleed control until blood lactate levels reached 4 mmol/L. During this variable hold window, AutoBleed continued to remove blood or reinfuse blood to maintain pressure at 35 mmHg. After the blood lactate hemorrhagic shock target was reached, animals received a calcium chloride bolus (1 g/10 mL) and were resuscitated using an adaptive resuscitation controller (ARC) with whole blood to a target MAP of 65 mmHg [20]. Whole blood was infused for 10 min using ARC to the target MAP, followed by switching the infusate to lactated Ringer’s (LR) solution for an additional 2 hours.

After the 2-hour resuscitation hold at target MAP was completed, animals underwent a re-bleed of identical magnitude and duration to the initial hemorrhage event. The animals were re-infused with LR using ARC to reach the target MAP and held for an additional 2 hours, followed by euthanasia with sodium pentobarbital (FatalPlus^®^).

### 2.2. Data Processing

For this study, only the baseline region through the whole blood resuscitation region was used in the data processing and ML model development methodology. MAP and volumetric blood hemorrhage and infusion data were recorded at 500 Hz and 1/5 Hz, respectively. The analog data were downsampled to a frequency of 100 Hz, and the digital data were upsampled to a frequency of 100 Hz for use in this study. The arterial waveform was filtered using a 512th-order finite impulse response (FIR) window lowpass filter with a cutoff frequency of 6 Hz. The pulse foot (diastolic trough), systolic peak, half-rise between the diastolic trough and systolic peak, the first inflection point, and the end of the waveform segment were calculated and identified for each waveform segment present in the arterial waveform [18]. In the absence of an inflection point of a waveform segment, the half-drop between the systolic peak and the following diastolic trough was calculated and identified [16]. The identified landmarks were then used to produce features using various mathematical manipulation techniques, based on previous research efforts [21,22,23,24], as well as features developed internally at the U.S. Army Institute of Surgical Research (USAISR), resulting in over 4200 features at each waveform segment of the arterial signal for each respective swine subject. In addition, a secondary set of features was extracted from an arterial waveform signal that was detrended using a fifth-order polynomial to eliminate baseline drift that may be present in the signal [25]. This detrending procedure removes the fluctuation of blood pressure seen by the pulse waves in the arterial waveform signal [26].

For the DL model, the analog data was downsampled, and the digital data was upsampled to a frequency of 100 Hz. This was done to match the sampling frequency of the training data used to create the DL model in previous works. To preprocess the data, it was segmented into intervals of 5 s, matching the pretraining data previously used to develop the DL model. No manual feature extraction was performed, as the neural network trains with convolutional neural network layers, which automatically extract features from the input signal [27,28].

### 2.3. Machine Learning Models

#### 2.3.1. Updates to Prior ML Prediction Models

The extracted features were used to develop ML models to predict calculated metrics for tracking both hemorrhage and resuscitation events in the animal study. Previous work developed blood loss metrics and quantified both time- and magnitude-sensitive hemorrhage metrics [16]. The three previously developed metrics were BLVM, PEBL, and HemArea. These metrics were designed to quantify different aspects of blood loss and its physiological impact. In brief, BLVM and PEBL quantified, on a 0–1 scale, how much blood was lost compared to the maximum volume of blood lost or the estimated blood volume of a swine, respectively. HemArea quantified blood loss over time by taking the area under the curve of a blood loss versus time plot. These prediction metrics are further outlined below.
(1)$$BLVM=1-\frac{Hemorrhaged\;Volume\;\left(t\right)}{Total\;Shed\;Hemorrhaged\;Volume}$$
(2)$$PEBL=\frac{Hemorrhaged\;Volume\;\left(t\right)}{Estimated\;Blood\;Volume\;Constant\;\frac{\mathrm{mL}}{\mathrm{kg}}\times Swine\;Weight\;\left(\mathrm{kg}\right)}$$
(3)HemArea=∑|BLVM−1|×∆t

In brief, BLVM was a 0–1 scale blood loss metric aimed at quantifying volumetric blood loss. PEBL quantified the percent blood loss in a subject based on standard blood volumes per weight constants. HemArea quantified the hemorrhage-time magnitude by summing area slices of BLVM and linearizing the results to estimate this magnitude.

Due to the inclusion of the resuscitation portion of the swine study, these original metrics based on blood loss had to be reworked to accurately reflect the desired prediction. BLVM was calculated using the original BLVM equation, but at the point of maximum blood loss, any infused whole blood volume was added instead of subtracted. This resulted in the blood balance “recovering”, as the metric no longer decreased and increased from the maximum volume of blood loss. The BLVM was still valued between 0 and 1, as the whole blood infusion volume necessary to resuscitate the swine back to the target MAP was never equal to the volume of whole blood hemorrhaged. However, its value could extend above one in studies where excess whole blood was infused.
(4)$$BLVM=1\;\pm \;\frac{Hemorrhaged\;Volume\;\left(t\right)\;or\;Whole\;Blood\;Infusion\;\left(t\right)}{Total\;Shed\;Hemorrhaged\;Volume}$$

The PEBL prediction required little modification from the original development in Gonzalez et al. [16]. The infusion of whole blood decreased the total blood loss of the swine by “giving back” the hemorrhaged whole blood, allowing the originally developed equation to be used. The subject’s estimated blood volume was changed from 80 mL/kg for canines to 60 mL/kg for swine [29].
(5)$$PEBL=\frac{Hemorrhaged\;Volume\;\left(t\right)}{60\;\frac{\mathrm{mL}}{\mathrm{kg}}\times Swine\;Weight\;\left(\mathrm{kg}\right)}$$

The HemArea prediction was previously calculated indirectly by taking slices of area under the BLVM curve, summing them, and performing a linear regression. This was originally done because the ML models performed poorly in predicting HemArea directly. Due to the addition of new features, a direct ML model was developed that could track HemArea, eliminating the need for the area under the curve process previously required.
(6)HemArea=∫Hemorrhage(t)dt

All prediction metrics developed in this study were smoothed using a moving mean with a window size of 50 data points to reduce noise and generalize trends. Each prediction underwent a linear regression vs. ground truth calculation to determine any data shifts, if necessary.

#### 2.3.2. Compensatory Reserve Measurement

The CRM underwent ML and DL developments in prior efforts [18,30]. These efforts define the compensatory reserve as the sum of all mechanisms of the body that act to protect against insufficient delivery of oxygen (DO_2_). To measure the compensatory reserve, a study was conducted where subjects were sealed inside a lower body negative pressure (LBNP) chamber, capable of redistributing blood volume from the upper body to the lower extremities due to vacuum pressure. This created a hypovolemic condition in the upper extremities of the participant inside the LBNP chamber. Increasing the vacuum pressure of the chamber in steps over time brought participants to the point of hemodynamic decompensation (HDD). This LBNP model allowed for CRM to be defined as follows:(7)$$CRM=1-\frac{LBNP}{LBNP_{HDD}}$$

LBNP is the vacuum pressure inside the chamber at a given point in time, and LBNP_HDD_ is the vacuum pressure in the chamber where the patient has reached HDD. This formula places CRM on a 0–100% scale, where 0% is the point of HDD and 100% places a patient at a full reserve. For the DL CRM model, a convolutional neural network (CNN) was developed as described in R.W. Techentin et al. [30]. Compared to ML models, a CNN does not require extracted features to be fed to it, as the convolutional layers with optimized hyperparameters, perform the feature extraction on their own. The CNN model was optimized with several hyperparameters in mind including but not limited to the number of layers, the number of filters, and kernel sizes. The optimization step led to the creation of a model with eight 1-D convolutional/pooling layers and other parameters provided in the publication from Techentin et al. The ML models developed to predict CRM consisted of a smaller pool of features than this study. The features were ranked using a ranking algorithm and then used as inputs to a bagged tree ML model for the prediction of CRM. The models in this study were pretrained using data that had ground truth CRM labels. Ground truth CRM could not to be defined, as the swine were not placed inside an LBNP chamber, and the point of HDD does not have a definition in this swine model.

#### 2.3.3. Machine Learning Model Development

Previous work used bagged tree ML models, as they were the highest-performing models compared in the study [18]. When retrained using the previously developed ML development framework (CRM-ML) for the current study, bagged tree ML models were compared to boosted tree ML models, and boosted trees performed better and were quicker to run. Prior work comparing boosted tree ML models to bagged tree ML models confirmed that boosted tree models provide better results in efficiency testing [31], which is important for real-time implementation of developed algorithms. The boosted tree ML model was the model chosen for the development of all ML models. Four groups of swine were created for the acquisition of the features used as inputs to the ML models. To prevent features from biasing towards a specific swine, three of the four groups of swine were concatenated, while leaving one group out. This was done four times to incorporate all the swine data. Each group’s features were ranked using the MRMR criterion in the MATLAB (v2023a, MathWorks, Natick, MA, USA) Regression Learner Toolbox. All the ML models were selected to use the top 20 features for consistency between the ML models developed from the different groups of swine data. Additionally, the boosted tree ML models all had a minimum leaf size of eight, a learning rate of 0.1, and went through 30 learning cycles. Once the features were obtained and the boosted tree models were trained with their respective groups of swine, they were tested using a cross-validation technique known as leave one subject out (LOSO) to account for bias in the testing. A swine group, which consisted of three swine subjects, was left out of the training process to blind test the ML model developed from the three other swine groups. This process was repeated four times in total, with different swine group combinations being used in the training, and the final swine group was split into individual swine subjects to be blind tested on each model (Figure 1). This resulted in a total of 12 LOSO processes being completed (four ML models, three blind tests each). The entire process was repeated for each prediction metric as well as on detrended data to identify if the ML models required overall trends in the signal for accurate predictions.

### 2.4. Machine Learning Model Analysis

After ML models were tuned for each application, their results were evaluated with blind subject holdouts (*n* = 3 subjects) for each LOSO model. Predictions were compared against ground truth calculations, except for the CRM models, as defining the point of decompensation in an anesthetized animal could not accurately be determined. Instead, the models were compared against MAP to provide some level of comparison. Model goodness of fit (R-Squared) and root mean squared error (RMSE) were used to compare predictions to ground truth calculations, and the results were averaged across all blind subject holdouts and LOSO models to obtain a generalized performance score. This was performed for the baseline, hemorrhage, and whole blood resuscitation datasets. Only trends were evaluated for the resuscitation phase, as after LR fluid was infused, the fluid balance would likely shift compared to whole blood based on characterized hemodynamic trends during hemorrhage resuscitation [32].

In addition, receiver operating characteristic curves (ROC) were calculated for the baseline and initial hemorrhage event to evaluate the performance of each model in accurately calculating hemorrhage status. The 95th and 5th percentiles for each model over this region were calculated using MATLAB to identify the range of possible values from each model. This range was subdivided into 100 threshold values for distinguishing hemorrhage and baseline regions, and true positive rate and false positive rate were calculated for each to generate the ROC curve. The area under the ROC (AUROC) was calculated for each model as well.

Models were further evaluated for the time taken to detect hemorrhage in each blind test subject. This was performed by identifying the 5th percentile for the baseline region of each dataset and determining at which time during the hemorrhage event this threshold value was reached for 100 consecutive readings, indicating a significant change from the baseline recording. This time was calculated for each predictive metric and MAP and averaged across all blind test subjects. Lastly, the effect of whole blood resuscitation on each metric was assessed by finding the average metric value in the 5 min prior to resuscitation compared to the final 5 min of the whole blood resuscitation event to quantify the change resulting from the resuscitation phase on each predictive metric as well as MAP.

Statistical analysis was performed to assess significant differences between each predictive model for AUROC, hemorrhage prediction time, and resuscitation responsiveness. This was done using Prism 10.3 (GraphPad, La Jolla, CA, USA). Normality was assessed by Shapiro-Wilk tests, and portions of each dataset were found to be non-normally distributed. As such, Friedman’s tests were used to compare datasets with the data from each swine paired across all metrics. A post-hoc Dunn’s test was used to compare CRM-DL, CRM-ML, BLVM, PEBL, HemArea, and MAP groups for AUROC, hemorrhage prediction time, and resuscitation responsiveness. *p*-values below 0.05 were considered significant, and a table of differences between each statistical test is summarized in Appendix A.

## 3. Results

This section details the development and results for each predictor ML model—BLVM, PEBL, HemArea, CRM—separately, followed by comparative performance results based on hemorrhage prediction and resuscitation tracking time.

### 3.1. Blood Loss Volume Metric Detrended vs. Non-Detrended Training

For the BLVM metric, ML models were developed in separate instances using non-detrended data and detrended data in a 12 LOSO cross-validation setup as described above. The models were then tested on both features extracted from the detrended and non-detrended arterial waveform to compare the average R-Squared and RMSE results. The ML models developed using non-detrended data had an R-Squared of 0.974 and 0.660 when tested on features extracted from non-detrended and detrended arterial waveforms, respectively. The RMSE values of the non-detrended ML models were 0.045 and 0.165 for non-detrended and detrended features, respectively. The ML models developed using detrended arterial waveform data had an R-Squared of 0.671 when tested on detrended features and 0.736 when tested on non-detrended features. The RMSE value of the models tested on detrended features was 0.162, while the RMSE value of the non-detrended features was 0.133. A summary of the BLVM R-Squared and RMSE metrics is shown in Table 1.

Blind prediction result differences were evident between using non-detrended and detrended datasets with the ML models developed using non-detrended data for BLVM, as shown in Figure 2. Figure 2a demonstrates the blind prediction of BLVM using a non-detrended ML model and non-detrended features and has a high correlation with the ground truth values in the baseline, hemorrhage, and hold regions, with a slight decrease in prediction accuracy occurring in the resuscitation region. Figure 2b demonstrates the blind prediction of BLVM using a non-detrended ML model and detrended features. These predictions tracked overall trends in the baseline, hemorrhage, hold, and resuscitation regions, but were less accurate overall when compared to the ground truth values. The overall correlation in all regions still exists, but the correlation was worse when removing overall trends in the data from which the features were extracted.

Blind prediction result differences were evident between using non-detrended and detrended datasets with the ML models developed using detrended data for BLVM, as shown in Figure 3. Figure 3a demonstrates the blind prediction of BLVM using the detrended ML model and detrended features. The model prediction has an overall correlation with the ground truth, though margins of error exist throughout, with an increase occurring in the resuscitation region, resulting in the features tracking with less accuracy in this region. Figure 3b demonstrates the blind prediction of BLVM using a detrended ML model with non-detrended features. The overall correlation between the predictions and the ground truth was still present, with margins of error increasing in the baseline and the resuscitation region, resulting in less accurate tracking in these regions.

### 3.2. Percent Estimated Blood Loss and Hemorrhagic Area

ML models to predict PEBL and HemArea were developed using non-detrended data to test with blind, features extracted from non-detrended data. The average of the 12 LOSO cross-validation setup resulted in R-Squared and RMSE values of 0.958 and 0.030, respectively, for the PEBL metric. The HemArea ML model had an R-Squared of 0.618 and a normalized RMSE of 0.703. A summary of the R-Squared and RMSE values for the PEBL and HemArea ML models can be seen in Table 2.

The ML models for predicting PEBL visually show a strong correlation when compared to the ground truth, as shown in Figure 4a. PEBL increases throughout hemorrhage and steadily drops once resuscitation starts. In Figure 4b, the predictions of the HemArea model correlated with overall trends, particularly in the baseline and the hold region, including the rapid oscillations during the hold. The HemArea predictions, however, have higher margins of error in the hemorrhage and resuscitation regions. This indicated that this metric was more challenging to predict compared to the other metrics, BLVM and PEBL, which had much stronger correlation scores.

### 3.3. Compensatory Reserve Metric

Both CRM models displayed a decrease in compensatory reserve as the swine subject underwent a hemorrhage event. The CRM-ML (Figure 5a) and CRM -DL (Figure 5b) models both flatline during the hold region, as expected, since there was no longer an active bleed occurring. There appeared to be an initial spike in both CRM models as the resuscitation was initiated, but this cannot be conclusive as the developed CRM models were trained only until the point of HDD, which does not include hemorrhagic shock resuscitation. Overall, both the CRM-ML and CRM-DL models perform as expected in the baseline through hold region of the study, demonstrating evidence that the physiological similarities of swine and humans may be applicable regarding CRM models previously developed. 

### 3.4. Model Predictive Performance Comparison

Next, overall performance for each metric was compared for early hemorrhage prediction time and their ability to track changes due to fluid resuscitation. We first evaluated each metric’s capability to distinguish non-hemorrhagic from hemorrhagic status during the initial hemorrhage event via ROC analysis (Figure 6a). Overall, BLVM had the strongest AUROC score at 0.998, with MAP and PEBL close behind at 0.979 and 0.951, respectively. The lower-performing AUROC scores were for CRM, with the DL and ML versions at 0.777 and 0.861, respectively, and HemArea at 0.850. BLVM and PEBL AUROC were significantly higher compared to CRM-DL, CRM-ML, and HemArea (summarized in Appendix A).

Hemorrhage prediction time was defined as when 100 straight readings were predicted in the 5th or 95th percentile of the baseline reading (Figure 6b). CRM-ML had the slowest prediction time at approximately 3000 s due to the metric often not reaching the prediction threshold during the hemorrhage window, resulting in it being scored at the maximum duration; all other metrics were significantly different from CRM-ML (summarized in Appendix A). The quickest predictions were with BLVM and PEBL at 78.8 and 82.0 s, respectively, both quicker than the traditional metric MAP at 116 s. We also evaluated how each predictive metric could be utilized for tracking the resuscitation phase of the experiment. This was initially defined in this study as a ratio of values immediately after resuscitation to immediately before resuscitation (Figure 6c). The strongest signal-to-noise ratio was for BLVM at 9.70, but the coefficient of variation across predictions was large at 109% (Figure 6d). However, BLVM was significantly larger than CRM-DL, CRM-ML, and PEBL (summarized in Appendix A). Conversely, MAP had the second highest ratio at 1.82 and the lowest coefficient of variation at only 10.3%.

## 4. Discussion

In both military and civilian trauma settings, timely definitive surgical control of hemorrhage and appropriate fluid resuscitation to restore circulating blood volume remain the most effective treatments for hemorrhagic shock, while delays to definitive care are associated with worse outcomes [33]. However, the appropriate application of these treatments requires the early and accurate identification of patients in impending hemorrhagic shock, which may not be readily apparent by traditional vital signs. Thus, reliance on traditional vital signs, such as blood pressure or heart rate, may delay the recognition of impending hemorrhagic shock and, consequently, delay definitive surgical and resuscitative treatment, leading to worse clinical outcomes. This could be especially important in the management of patients with internal bleeding, such as hemothorax and intraabdominal-hemorrhage, where ongoing blood loss may not always be readily apparent to clinicians. Additionally, traditional vital signs may fail to accurately stratify patients by acuity, leading to incorrect triaging of patients in mass casualty incidents when prioritizing patients for care or evacuation is critical. Accordingly, we propose that these novel metrics may be able to more accurately identify critically ill patients earlier than traditional vital signs, thus allowing the appropriate selection of patients for intervention and the earlier application of appropriate treatments, improving clinical outcomes compared to care that relies on traditional vital signs.

For this primary motivation, we evaluated different ML approaches to develop methods to track metrics that can measure hemorrhagic blood volume loss and resuscitation during a hemorrhage, as well as detect the occurrence of a hemorrhagic event earlier than traditional vital signs using a swine-controlled hemorrhage model. By tracking both hemorrhage and resuscitation, these advanced metrics could simultaneously assist in the accurate diagnosis and targeted treatment of hemorrhagic shock, leading to better overall patient outcomes. Two approaches were taken for this study—(i) evaluate previously developed blood loss-based metrics using these new swine hemorrhage datasets and (ii) develop ML models tuned for the swine hemorrhage and resuscitation datasets.

The previously developed ML feature selection framework was utilized to develop models for the prediction of BLVM, PEBL, and HemArea, which were validated using a LOSO cross-validation setup. This was critical due to the limited dataset size to conduct blind subject testing with the maximum amount of noise to ensure models were not overfitting noise and data artifacts. Overall, each metric showed success in tracking the onset of hemorrhage as well as the resuscitation that followed. BLVM and PEBL showed a higher goodness of fit metric between the ML prediction and calculated ground truth. HemArea performed worse by comparison to these metrics but still generally tracked the experimental phases. We previously calculated HemArea from BLVM; in this effort, we predicted HemArea [16]. The reduced performance of this metric may be due to this difference, and, thus, calculating of HemArea from BLVM may be a more suitable prediction approach. Similarly, BLVM and PEBL provided earlier prediction time compared to MAP, a more traditional metric for tracking hemorrhage onset, but HemArea took slightly longer to detect hemorrhage. It is worth noting that all metrics provided much earlier hemorrhage detection compared to results from our prior canine hemorrhage model (i.e., 12 min in canines vs. 82 s in swine for PEBL). This is likely due to the splenectomy performed in swine prior to hemorrhage, which reduced physiological compensation and was not performed in canines. The canines likely had a larger compensatory reserve at the onset of hemorrhage, allowing for more effective masking of an impending hemorrhage event.

Further work will be required to increase the correlation of HemArea to its ground truth, likely by adding features that tend to correlate with HemArea based on the current top-ranking features for HemArea. One approach will be to calculate HemArea from BLVM predictions, as the BLVM predictive models had strong correlation scores to derive HemArea more accurately as we have previously done [16]. In addition, spectral features have been used for the estimation of cardiovascular parameters [34] and are currently not used in the ML model development for this work. Adding these features may further improve ML models for directly predicting HemArea. Future development of ML models to track blood loss and resuscitation would entail using non-invasively obtained physiological data, such as a photoplethysmography (PPG) waveform, as input to the feature extraction framework developed in previous studies. Due to the PPG signal having fewer or no overall trends like the arterial waveform data, the results from the metrics obtained from the detrended arterial waveform may provide insight into the feasibility of transferring this feature extraction framework into non-invasive methods. However, as observed in the results, models that used detrended data generally performed worse than models using non-detrended data. A range of limitations could have caused this shortcoming, including a lack of subject variability and model complexity. This study utilized 12 different swine subjects, where only 9 were used for training in the cross-validation setup. There is reason to believe that 9 subjects were not sufficient for the model to generalize the data enough to accurately track blind data. This issue has a straightforward solution—obtaining more datasets from multiple different swine subjects—but protocols to do so are costly and time-consuming.

We also evaluated the use of previously developed metrics for compensatory reserve measurement. CRM was trained using hundreds of human subjects experiencing simulated hemorrhage through LBNP exposure [35]. The original CRM model uses a DL framework, but we have also previously evaluated the use of feature extraction and decision tree models to track CRM in subjects undergoing an LBNP procedure to simulate central hypovolemia [18]. Due to the similarities in physiology between humans and swine, we took both the DL and ML models trained using human data and made blind predictions on swine datasets. This was also done because the point of decompensation, needed for training the CRM model, cannot be readily defined in an anesthetized animal. While CRM models showed trends for tracking hemorrhage onset, the results were more variable across each swine subject, with lower signal-to-noise.

Further steps to address the limitations of this study would include addressing model complexity. The boosted tree models were chosen for their efficiency, with the prospect that they would be able to operate in real time. However, to effectively track blood loss, DL may prove to be a more worthwhile approach. The boosted tree models require feature extraction to be performed on the data, which led to the development and ranking of the top 20 features used in this study. While those top 20 features originated from a 4200+ feature pool it is possible that an even greater feature pool exists, but this would require a higher degree of expertise to derive additional features. This may lead to the use of CNNs, such as the CRM DL model, for future development. The nature of performing convolutions on a piece of data allows the model to extract features itself, and when trained adequately, the model can rank the features necessary to track the data accurately. While CNN models can certainly reach levels of complexity that could make them computationally burdensome and unlikely to be used in real time, the power of even a single convolutional layer has the potential to extract the necessary features from a waveform to make accurate predictions.

Finally, there are some limitations in using swine as a research model for humans. While swine share many physiological similarities with humans, they are still a different species, leading to subtle but important differences. Since machine learning algorithms are something of a “black box”, it is not immediately clear what aspects of swine physiology the algorithms are using to determine blood pressure, and it is therefore unclear whether the algorithm could make the leap from one species to another. Still, large animal research provides an avenue for robustly demonstrating proof of concept for these machine learning approaches, such that even if the algorithms themselves cannot translate directly from one species to another, the methods used to develop those algorithms should be able to translate between species.

## 5. Conclusions

In conclusion, ML algorithms developed for tracking blood loss during hemorrhage were successfully created using a swine model, whose physiology is similar to humans, with accurate correlations and earlier prediction times compared to traditional vital signs. These metrics allowed for earlier, more consistent prediction of hemorrhage compared to traditional metrics such as blood pressure. If improved blood loss prediction metrics can be established for hemorrhage and resuscitation, surgical intervention to control hemorrhage and goal-directed fluid resuscitation to restore volume status could be administered earlier than current gold standards, improving patient outcomes during civilian trauma and combat casualty care.

## 6. Patents

J.M.G. and E.J.S. are co-inventors on a filed provisional patent owned by the U.S. Army related to the blood loss volume metric and other similar predictive models (filed 14 August 2023).

## Figures and Tables

**Figure 1 bioengineering-11-01075-f001:**
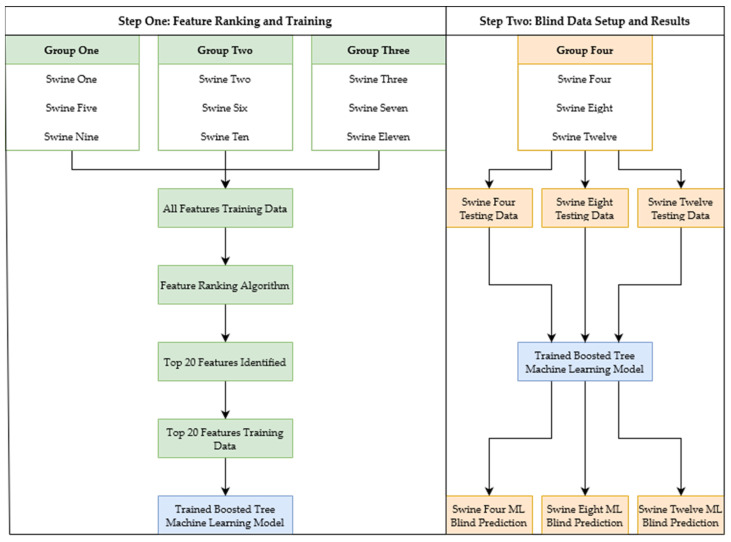
The ML development pathway for one of the four ML models. The ML training data was split, resulting in an ML model (blue) after undergoing the training pipeline. The ML model (blue) is then input into the testing stage, resulting in ML model predictions for three swine completely left out of the training stage. The process displayed is repeated a total of four times with different arrangements of the four groups, with each group completely left out for testing once.

**Figure 2 bioengineering-11-01075-f002:**
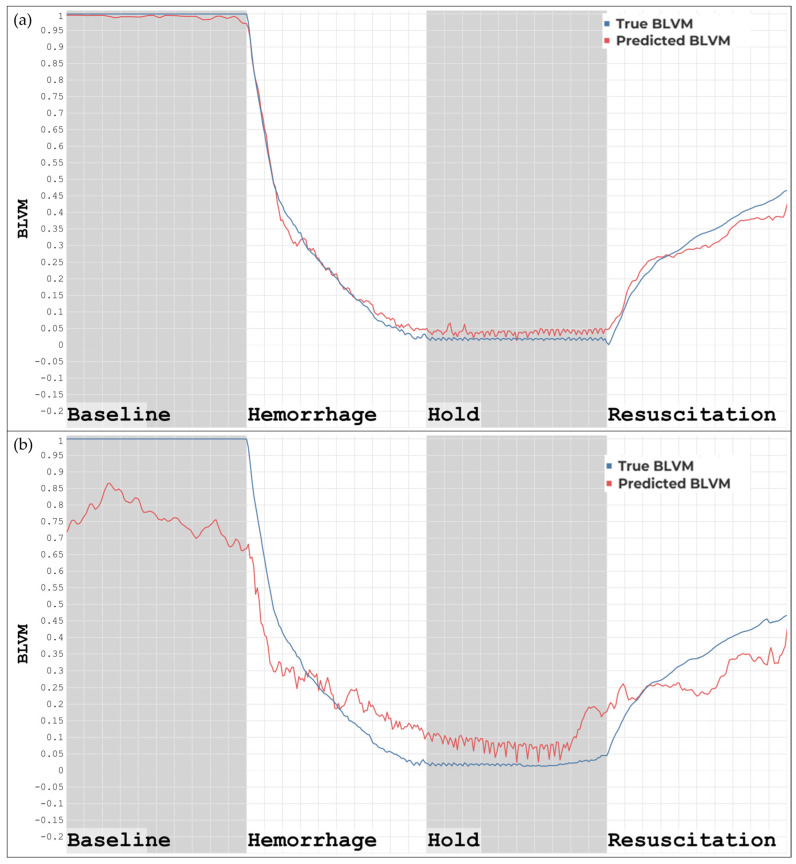
(**a**) ML model developed using non-detrended data tested on features extracted from non-detrended data. (**b**) ML model developed using non-detrended data tested on features extracted from detrended data.

**Figure 3 bioengineering-11-01075-f003:**
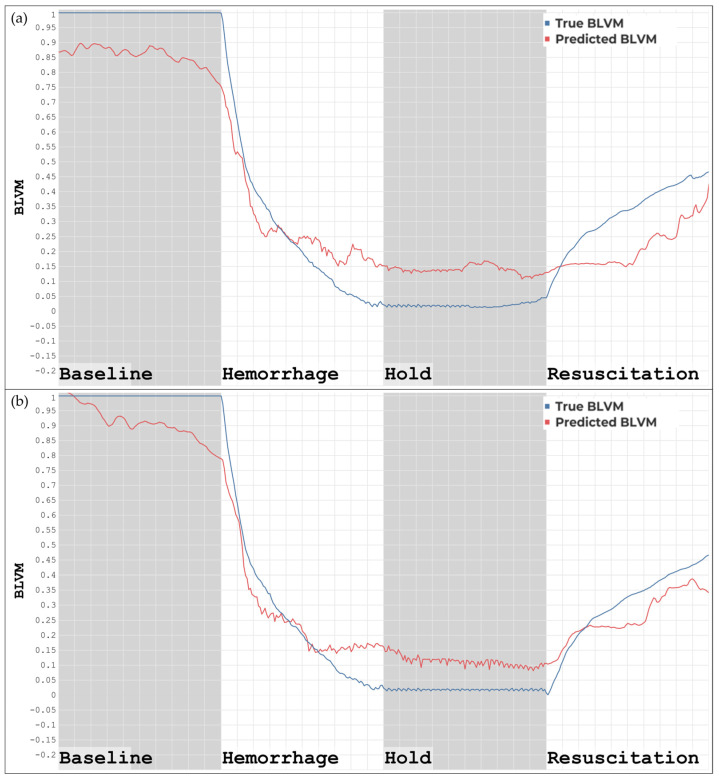
(**a**) ML model developed using detrended data tested on features extracted from detrended data. (**b**) ML model developed using detrended data tested on features extracted from non-detrended data.

**Figure 4 bioengineering-11-01075-f004:**
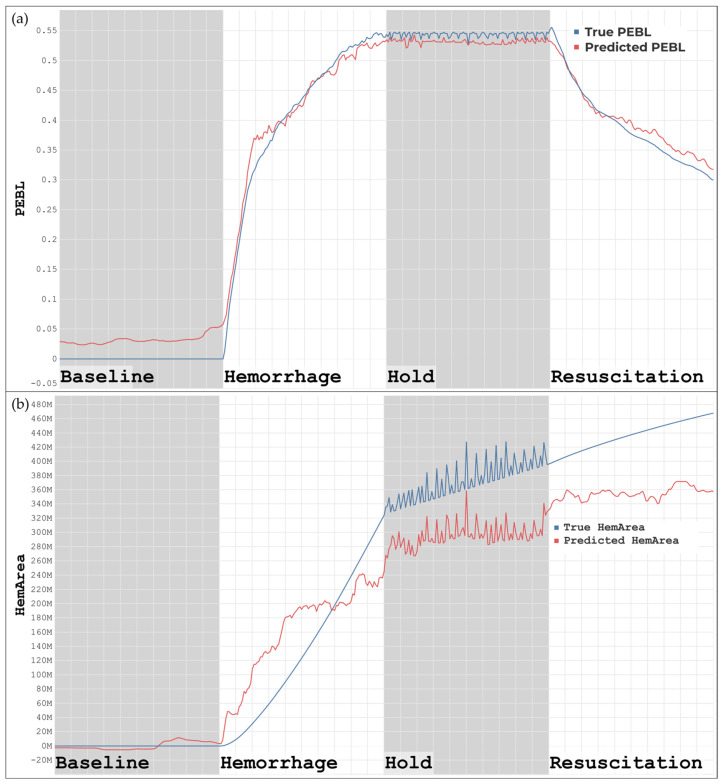
(**a**) ML model developed using unfiltered data for the prediction of PEBL. (**b**) ML model developed using unfiltered data for the prediction of HemArea.

**Figure 5 bioengineering-11-01075-f005:**
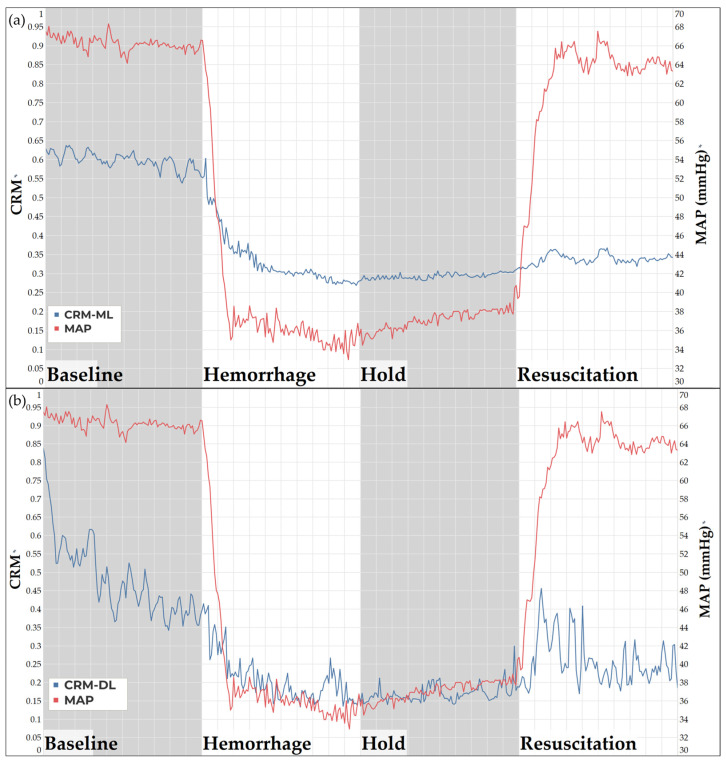
(**a**) ML model trained on human data, tested on swine for CRM predictions. (**b**) DL model trained on human data, tested on swine for CRM predictions.

**Figure 6 bioengineering-11-01075-f006:**
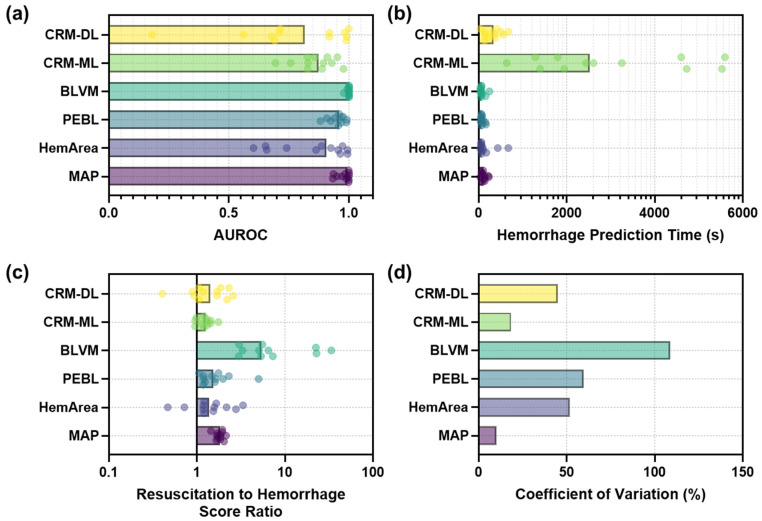
Comparative performance of each predictive model for (**a**) Area under the ROC curve, (**b**) Time to consistent hemorrhage prediction (note: logarithmic scale on *y*-axis), and (**c**) Ratio of metric value after resuscitation to during hemorrhage to create a signal-to-noise metric (note: logarithmic scale on *y*-axis). (**d**) Comparison of coefficient of variation (%) for the signal-to-noise resuscitation to hemorrhage ratios across each trained model.

**Table 1 bioengineering-11-01075-t001:** Average R-Squared and RMSE values from ML models developed using blind non-detrended and detrended waveforms on features extracted from non-detrended and detrended waveforms.

Non-Detrended ML Model, Non-Detrended Features		Non-Detrended ML Model, Detrended Features		Detrended ML Model, Detrended Features		Detrended ML Model, Non-Detrended Features	
R-Squared	RMSE	R-Squared	RMSE	R-Squared	RMSE	R-Squared	RMSE
0.974	0.045	0.660	0.165	0.671	0.162	0.736	0.133

**Table 2 bioengineering-11-01075-t002:** Average R-Squared and RMSE values across 12 LOSOs of PEBL and HemArea predictions versus their respective ground truth.

PEBL		HemArea	
R-Squared	RMSE	R-Squared	RMSE *
0.958	0.030	0.618	0.703

* Normalized by the max value obtained during the LOSO process.

## Data Availability

The data presented in this study are not publicly available because they have been collected and maintained in a government-controlled database located at the U.S. Army Institute of Surgical Research. This data can be made available through the development of a Cooperative Research and Development Agreement (CRADA) with the corresponding author.

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
