# Peer review of "Machine Learning Models for Tracking Blood Loss and Resuscitation in a Hemorrhagic Shock Swine Injury Model"

_bioengineering, 2024, doi:10.3390/bioengineering11111075_

Round 1

Reviewer 1 Report

Comments and Suggestions for Authors

An interesting manuscript investigating the role of AI for tracking blood loss and resuscitation in a hemorrhagic shock. The authors present signal processing methods and feature extraction techniques to identify hidden patterns and characteristics within physiological signals and eventually aim to develop more sensitive and earlier indicators of hemorrhagic shock. This work is based on swine model.

The manuscript is well written and presented and appropriate for the journal. Has an introduction containing the motivation, describing what is feature extraction and machine learning, and furthermore presents existing work and the study aim. It is followed by a detailed materials and methods section. The results are presented for each machine learning model developed by the authors and is followed by comparative performance on the basis of hemorrhage prediction and resuscitation tracking time. Finaly, is the discussion and conclusions.

There are only a few comments that may help to improve this work:

1.       “Machine learning (ML) and deep learning (DL) are subsets of AI that focus on building algorithms capable of learning and making predictions on data” actually ML algorithms are not only for prediction, perhaps some rephrasing here can make this sentence more accurate.

2.       Lines 145 and 146, it is not provided information for the 8th animal (n=8 as mentioned in line 142). Furthermore it is confusing since later (as figure 1 clearly depicts) is clear that the are 12 animals. Propose to clarify at this point the total number of animals of the study and their grouping.

3.       “Analog and digital data were recorded at 500 Hz and 1/5 Hz, respectively. The analog data was downsampled to a frequency of 100 Hz and the digital data was upsampled to a frequency of 100 Hz for use in this study.”. In my opinion this is the place to report what is the acquired analog and the digital data, perhaps a table?

Author Response

An interesting manuscript investigating the role of AI for tracking blood loss and resuscitation in a hemorrhagic shock. The authors present signal processing methods and feature extraction techniques to identify hidden patterns and characteristics within physiological signals and eventually aim to develop more sensitive and earlier indicators of hemorrhagic shock. This work is based on swine model.

The manuscript is well written and presented and appropriate for the journal. Has an introduction containing the motivation, describing what is feature extraction and machine learning, and furthermore presents existing work and the study aim. It is followed by a detailed materials and methods section. The results are presented for each machine learning model developed by the authors and is followed by comparative performance on the basis of hemorrhage prediction and resuscitation tracking time. Finaly, is the discussion and conclusions.

There are only a few comments that may help to improve this work:

  1. “Machine learning (ML) and deep learning (DL) are subsets of AI that focus on building algorithms capable of learning and making predictions on data” actually ML algorithms are not only for prediction, perhaps some rephrasing here can make this sentence more accurate.

Thanks for taking the time to review our submitted manuscript. We agree with the reviewer and have modified the sentence to say that “…are subsets of AI that can focus on building…” to make it more evident that ML/DL apply to other applications as well.

  1. Lines 145 and 146, it is not provided information for the 8thanimal (n=8 as mentioned in line 142). Furthermore it is confusing since later (as figure 1 clearly depicts) is clear that the are 12 animals. Propose to clarify at this point the total number of animals of the study and their grouping.

We agree that further clarification was needed regarding the anesthetic used. In summary, 8 swine used isoflurane throughout the study, 3 used ketamine, and 1 used propofol. These amount to the 12 swine used to develop the AI models in this study. Further, based on other reviewer comments, we have decided to remove this detail on anesthetics used as this work did not focus on the development of AI-models specific to different anesthetics used and the continued inclusion of these details may add unnecessary confusion to a reader.

  1. “Analog and digital data were recorded at 500 Hz and 1/5 Hz, respectively. The analog data was downsampled to a frequency of 100 Hz and the digital data was upsampled to a frequency of 100 Hz for use in this study.”. In my opinion this is the place to report what is the acquired analog and the digital data, perhaps a table?

Thank you for pointing this out. While a range of data were recorded in this study, we only used mean arterial pressure analog data and volumetric blood loss/infusion volumes digital data. We have added clarification to reflect this in the manuscript.

Reviewer 2 Report

Comments and Suggestions for Authors

Q1: This paper is interesting. The authors have tried to use machine learning for early hemorrhage detection and resuscitation tracking is scientifically valid and well-grounded. The study successfully applies advanced signal processing techniques, and the methodology is well-described. However, the abstract should clearly explain how the machine learning metrics contribute to practical medical outcomes. Also, the authors should consider expanding on the implications of using swine models for human medical practice in terms of limitations or expected next steps. The discussion is scientifically sound, but a deeper analysis of the limitations—particularly regarding the generalization of results from swine to humans—would enhance the article's completeness.

Q2: Abstract: line 28

Causality:" is an influence by which one event, process, state, or object (a cause) contributes to the production of another event, process, state, or object 

Casualty:  damage

I am confused by your causality here

Q3: line 97: Hypovolemic shock often present” themselves"   should be singular to match “shock.”

Q4: Introduction : The introduction effectively describes the problem of hemorrhagic shock and motivates the study. However, it could benefit from a clearer linkage between the physiological background and the machine learning approach to ensure smooth reading.

Q5: Machine Learning Models:  the section dealing with detrended and non-detrended data comparison should further clarify why detrending was necessary in the first place and its practical significance.

Q6: Materials and Methods: The description of the swine model is thorough, but the inclusion of analgesia and anesthesia methods  might overwhelm the reader with unnecessary detail. Consider condensing these technical descriptions into a short paragraph unless critical for machine learning feature extraction.

Q7: Materials and Methods: Controlled hemorrhage procedure: When describing the AutoBleed control and MAP targets, providing a brief rationale for choosing those specific thresholds (e.g., why 35 mmHg?) could help clarify the clinical relevance.

Q8: Materials and Methods:  Machine Learning Models :explaining why DL was chosen for some models (e.g., automatic feature extraction in convolutional neural networks) versus more traditional methods might help readers unfamiliar with ML/DL understand its necessity.

Q9: Materials and Methods:  Comparison of metrics: Introducing a quick comparison of the three key blood loss metrics (BLVM, PEBL, HemArea) early in this section would provide a useful roadmap for the reader. For example, after introducing BLVM, you could add: "These metrics were designed to quantify different aspects of blood loss and its physiological impact, as outlined below."

Q10: Materials and Methods:  Statistical significance: More information on the statistical methods used to validate the ML models would be beneficial, such as whether p-values or confidence intervals were calculated to assess model performance.

Q11: Results, Blood Loss Metrics : Adding more direct explanations to accompany Figures 2 and 3 would help the reader understand their significance. For example: "As shown in Figure 2a, the non-detrended ML model closely tracks the true values in the baseline and hemorrhage regions, but its accuracy decreases slightly during resuscitation."

Q12: Percent Estimated Blood Loss and Hemorrhage Area (Section 3.2). The explanation of the PEBL model is sound, but the HemArea model’s lower performance should be discussed in more detail. Why is the correlation lower for HemArea? Is it due to inherent challenges in predicting hemorrhage area, or are there specific model limitations?

Q13: Figure 4, you could add a brief summary: "The ML models for PEBL demonstrate a high correlation with ground truth, particularly during hemorrhage events, as seen in Figure 4a, whereas HemArea (Figure 4b) shows higher variance during these phases."

Q14: Discussion: While the study does a good job of detailing how the machine learning models perform, more emphasis on how these results translate into practical clinical improvements would strengthen the discussion. For instance, how might these early detection metrics reduce mortality in military or civilian contexts? Including more specific applications would enhance the paper’s impact.

Q15: Limitations: The paper briefly touches on limitations, but a more detailed discussion of the constraints in using swine models for human applications is necessary. For example, swine physiology is similar but not identical to human physiology—how might this affect the generalizability of the results?

Q16: Comparison with Existing Models: Briefly comparing the ML and DL models developed here with existing traditional methods would help contextualize the improvements. Adding a sentence "In contrast to traditional vital sign-based models, which detect hemorrhage late in the process, our ML models detect blood loss as early as 82 seconds into the hemorrhage event" would emphasize the contribution.

Q17: Conclusion: While the conclusion succinctly summarizes the study’s results, reinforcing the significance of early detection and how the ML models outperform traditional methods would be impactful. For example, re-emphasizing that the early prediction capabilities of these models could lead to faster interventions would help underline the practical implications.

Q18: Future Directions: The paper could benefit from a more specific statement on future research directions. For example, how will the authors improve the HemArea predictions? What steps are planned for validating these models in human clinical settings?

Q19: Being a clinician, I would expect the authors can expand their discussion of how the results might directly translate to improving patient outcomes in both military and civilian trauma settings. 

Author Response

This paper is interesting. The authors have tried to use machine learning for early hemorrhage detection and resuscitation tracking is scientifically valid and well-grounded. The study successfully applies advanced signal processing techniques, and the methodology is well-described. However, the abstract should clearly explain how the machine learning metrics contribute to practical medical outcomes. Also, the authors should consider expanding on the implications of using swine models for human medical practice in terms of limitations or expected next steps. The discussion is scientifically sound, but a deeper analysis of the limitations—particularly regarding the generalization of results from swine to humans—would enhance the article's completeness.

  1. Abstract: line 28; Causality:" is an influence by which one event, process, state, or object (a cause) contributes to the production of another event, process, state, or object. I am confused by your causality here

We thank the reviewer for carefully reviewing our manuscript. This was a typographical error that bas been corrected to read casualty instead.

  1. Line 97: Hypovolemic shock often present” themselves"   should be singular to match “shock.”

We have re-worded this sentence to read: “Unfortunately, symptoms of hypovolemic shock often become apparent when life-saving interventions are less effective.” We hope this edit makes the sentence read more clearly.

  1. Introduction : The introduction effectively describes the problem of hemorrhagic shock and motivates the study. However, it could benefit from a clearer linkage between the physiological background and the machine learning approach to ensure smooth reading.

Thank you for your feedback. The introduction was amended to include information to better transition from the physiological background to machine learning, as well as a description of how physiological data can be used to train machine learning models.

  1. Machine Learning Models:  the section dealing with detrended and non-detrended data comparison should further clarify why detrending was necessary in the first place and its practical significance.

Additional information on why detrending was done was incorporated into the methodology sections, and the future significance was discussed in the future work paragraph of the conclusion.

  1. Materials and Methods: The description of the swine model is thorough, but the inclusion of analgesia and anesthesia methods  might overwhelm the reader with unnecessary detail. Consider condensing these technical descriptions into a short paragraph unless critical for machine learning feature extraction.

Thank you for the insight. After consideration, we have removed the details as they are not part of the machine learning feature extraction (e.g., ML models were not developed for specific anesthetics). We are required by IACUC policy to mention that swine were kept under a surgical plane of anesthesia, but other details were removed to simplify the manuscript.

  1. Materials and Methods: Controlled hemorrhage procedure: When describing the AutoBleed control and MAP targets, providing a brief rationale for choosing those specific thresholds (e.g., why 35 mmHg?) could help clarify the clinical relevance.

This was chosen to be sufficiently reduced in blood volume to achieve hypovolemic shock based on experience with swine models but confirmed via blood lactate readings. Slightly more details were provided to help clarify.

  1. Materials and Methods:  Machine Learning Models :explaining whyDL was chosen for some models (e.g., automatic feature extraction in convolutional neural networks) versus more traditional methods might help readers unfamiliar with ML/DL understand its necessity.

Thanks for the suggestion.  Our reasoning for having a DL model is that the model was already trained and provided from a prior effort. This is pointed out in section 2.2.2.

  1. Materials and Methods:  Comparison of metrics: Introducing a quick comparison of the three key blood loss metrics (BLVM, PEBL, HemArea) early in this section would provide a useful roadmap for the reader. For example, after introducing BLVM, you could add: "These metrics were designed to quantify different aspects of blood loss and its physiological impact, as outlined below."

Thank you for the feedback. We have incorporated the suggestions into the manuscript as well as added a brief description of the metrics prior to the more in-depth analysis presented in the work.

  1. Materials and Methods:  Statistical significance: More information on the statistical methods used to validate the ML models would be beneficial, such as whether p-values or confidence intervals were calculated to assess model performance.

Statistical analysis was not initially performed but we agree with the reviewer and have details across the manuscript, including methodological description in the methods, highlighting significant differences in the results, and included significant differences in supplementary tables due to the complexity of the analysis. Figures have been updated to reflect these changes as well. 

  1. Results, Blood Loss Metrics : Adding more direct explanations to accompany Figures 2 and 3 would help the reader understand their significance. For example: "As shown in Figure 2a, the non-detrended ML model closely tracks the true values in the baseline and hemorrhage regions, but its accuracy decreases slightly during resuscitation."

Thank you for your feedback. We have edited the manuscript to add additional clarity to the sections discussed.

  1. Percent Estimated Blood Loss and Hemorrhage Area (Section 3.2). The explanation of the PEBL model is sound, but the HemArea model’s lower performance should be discussed in more detail. Why is the correlation lower for HemArea? Is it due to inherent challenges in predicting hemorrhage area, or are there specific model limitations?

We have added slightly more details in the results but have further expanded on the challenges with HemArea prediction model in the more appropriate discussion section.

  1. Figure 4, you could add a brief summary: "The ML models for PEBL demonstrate a high correlation with ground truth, particularly during hemorrhage events, as seen in Figure 4a, whereas HemArea (Figure 4b) shows higher variance during these phases."

              We have added a summary statement as described above in the results section.

  1. Discussion: While the study does a good job of detailing how the machine learning models perform, more emphasis on how these results translate into practical clinical improvements would strengthen the discussion. For instance, how might these early detection metrics reduce mortality in military or civilian contexts? Including more specific applications would enhance the paper’s impact.

We appreciate the opportunity to expand the discussion to include information about the clinical relevance of the novel metrics in the early identification of patients in need of definitive surgical management and how that might translate to better clinical outcomes.

  1. Limitations: The paper briefly touches on limitations, but a more detailed discussion of the constraints in using swine models for human applications is necessary. For example, swine physiology is similar but not identical to human physiology—how might this affect the generalizability of the results?

Thank you for this question and suggestion: we have included a brief discussion of the applicability of swine models to human clinical care.

  1. Comparison with Existing Models: Briefly comparing the ML and DL models developed here with existing traditional methods would help contextualize the improvements. Adding a sentence "In contrast to traditional vital sign-based models, which detect hemorrhage late in the process, our ML models detect blood loss as early as 82 seconds into the hemorrhage event"would emphasize the contribution.

Comparison to traditional metric is MAP or blood pressure which is emphasized in the discussion sections. We have further expanded on this point in the results section.

  1. Conclusion: While the conclusion succinctly summarizes the study’s results, reinforcing the significance of early detection and how the ML models outperform traditional methods would be impactful. For example, re-emphasizing that the early prediction capabilities of these models could lead to faster interventions would help underline the practical implications.

As suggested, we have added additional details in the conclusion section. 

  1. Future Directions: The paper could benefit from a more specific statement on future research directions. For example, how will the authors improve the HemArea predictions? What steps are planned for validating these models in human clinical settings?

Thank you for your feedback. Regarding the improvements of ML models for prediction of HemArea, we have added a description of additional features in the discussion section that  are currently not in our analysis, but can be added and may correlate and improve, not only HemArea, but all the ML model predictions of each metric.

  1. Being a clinician, I would expect the authors can expand their discussion of how the results might directly translate to improving patient outcomes in both military and civilian trauma settings. 

Thank you for the suggestion: we have included more information about how the results of the study could translate into improved patient outcomes.